# How do GPs and patients share the responsibility for cancer safety netting follow-up actions? A qualitative interview study of GPs and patients in Oxfordshire, UK

Julie Evans,[1] John I Macartney,[1] Clare Bankhead,[1] Charlotte Albury,[1] Daniel Jones,[2] Sue Ziebland,[1] Brian D Nicholson[1]

[1]Nuffield Department of Primary Care Health Sciences, University of Oxford, Oxford, UK
[2]Supportive care, early diagnosis and advanced disease research group, Hull York Medical School, Hull, UK

**Correspondence to**
Dr Brian D Nicholson;
brian.nicholson@phc.ox.ac.uk

## ABSTRACT

**Objective** To explore patients' and General Practitioners' (GPs) accounts of how responsibility for follow-up was perceived and shared in their experiences of cancer safety netting occurring within the past 6 months.

**Design** In-depth interviews were recorded and transcribed verbatim. Data were analysed through an abductive process, exploring anticipated and emergent themes. Conceptualisations of 'responsibility' were explored by drawing on a *transactional* to *interdependent* continuum drawing from the shared decision-making literature.

**Settings and participants** A purposive sample of 25 qualified GPs and 23 adult patients in Oxfordshire, UK.

**Results** The transactional sharing approach involves responsibility being passed from GP to patient. Patients expected and were willing to accept responsibility in this way as long as they received clear guidance from their GP and had capacity. In interdependent sharing, GPs principally aimed to reach consensus and share responsibility with the patient by explaining their rationale, uncertainty or by stressing the potential seriousness of the situation. Patients sharing this responsibility could be put at risk if no follow-up or timeframe was suggested, they had inadequate information, were falsely reassured or their concerns were not addressed at re-consultation.

**Conclusion** GPs and patients exchange and share responsibility using a combination of transactional and interdependent styles, tailoring information based on patient characteristics and each party's level of concern. Clear action plans (written where necessary) at the end of every consultation would help patients decide when to re-consult. Further research should investigate how responsibility is shared within and outside the consultation, within primary care teams and with specialist services.

## INTRODUCTION

There is a growing literature examining why patients consult their doctor with symptoms that could represent cancer[1]; yet there is almost no evidence about how General Pratitioners (GPs) engage with symptomatic

### Strengths and limitations of this study

► We report the first study to explore how General Practitioners (GPs) and patients handover or share responsibility for follow-up actions in the context of possible cancer.
► We conducted interviews within 6 months of a cancer safety netting episode to reduce the risk of participant recall bias.
► We achieved a varied purposeful sample of GPs and patients although a wider geographical and sociodemographic sample may have illuminated additional issues.
► Self-selected GP participants may have had a stronger commitment to safety netting than those who declined to take part.

patients once they have attended.[2] Safety netting is a strategy used to ensure that patients presenting with symptoms or signs are monitored until they have resolved or an explanation is reached.[3] Several components of safety netting have been described including imparting information and advice about what to expect and when to re-consult, reminding patients to re-consult appropriately, and following up and acting on test results.[4] Initially conceptualised as a milestone of communication within the consultation, safety netting has been developed to incorporate wider clinician behaviours and health system functions.[5 6]

Shared decision-making (SDM) is widely discussed as an interdependent process in which health professionals, patients and their caregivers relate to and influence each other in making decisions about a patient's health.[7] SDM occupies a middle ground between the 'paternalist' doctor and the 'autonomous' patient, with varied interpretations of the

breadth of interactions taking place in contemporary clinical practice.[8–11] While wide consideration has been given to the policy and practices of SDM[12 13] a low quality evidence base perpetuates uncertainty about which of the many components of SDM are most effective.[14] When SDM is based on a synthesis of patient's desires, values and preferences it is argued that it can enhance communication, understanding and signal respect.[15] Limited time, doctor and patient preference, and ineffective communication are offered as explanations for why SDM has not easily translated into routine practice.[9] Attention has focused on the benefits of advocating patient choice and exploring when clinicians might act (paternalistically) in a patient's best interest.[16–19] Less consideration has been given to the limitations of patient autonomy and in what circumstances patients prefer to make their own decisions.[20–23]

Little is known about how patients and GPs negotiate responsibility for safety netting when cancer is a possible diagnosis. Qualitative research has demonstrated that cancer patients may feel abandoned if their doctors appear to be leaving a potentially life changing decision entirely in the patient's court.[24] Consensus safety netting guidance aims to ameliorate this by stating that GPs should give specific information about when and how best to re-consult, including who has the responsibility to make the appointment.[3] Our own work has shown that GPs report relying on competent patients to take the responsibility to act on that advice to re-consult once they have explained their thinking and expectations, while being more proactive in arranging follow-up for patients they perceive to be at higher risk or less able to take responsibility.[25] Building on these findings, the aim of this paper is to explore patients' and GPs' accounts of how responsibility was perceived and shared in their own recent experiences of cancer safety netting.

## METHODS

### Recruitment

We advertised the study to GPs in Oxfordshire via the Clinical Research Network and local clinical commissioning group. GPs known through our institution and those expressing interest from the advert were sent an invitation letter, information sheet and reply slip. We aimed for maximum variation in age, length of time in practice and rural or urban setting, and for saturation in our major analytic categories. After interview, participating GPs were asked to pass an information pack to one or two patients who they considered had been 'safety netted' for possible cancer within the previous 6 months. These patients had not been referred on an urgent cancer pathway at the first consultation, and had either been diagnosed with cancer or had it ruled out. Patient recruitment was supplemented by placing advertisements in GP surgeries, cancer support centres, patient involvement websites, community information websites, and on the study webpage. Those who responded were sent an information pack.

### Interviewing

We developed a series of interview prompts from our knowledge of the literature, secondary analyses of surveys related to safety netting and our recent qualitative study of cancer diagnosis.[4 26] Participants were contacted by JE—a female qualitative social scientist specialising in patient experiences of cancer—to arrange an interview at a time and place to suit the participant, either their home, the researcher's office or the GP's surgery (GP interviews only). Written informed consent was obtained. To avoid constraining the accounts we intentionally avoided defining the term 'safety netting' at the outset allowing the participant to explore it using their own words. In addition to eliciting their experiences and views of safety netting, all participants were asked for their opinion on whose responsibility it should be to make sure that patients returned for a follow-up visit. From this starting point, JE developed the 'responsibility' theme cycling between the conduct of interviews and preliminary analysis.[27] Interviews lasted an hour on average, were digitally recorded and transcribed verbatim.

### Analysis

NVivo V.10 qualitative data analysis software was used to code the transcripts to anticipated and emergent themes using constant comparison, a method for ensuring all aspects of the data are considered.[28] Analysis for this article was led by JE and JIM, in discussion with other co-authors. Data on the rich theme of 'responsibility' were examined by members of the research team (JE, JIM, CB, BDN, SZ) using the one sheet of paper method, a qualitative mind-mapping approach to analysis.[29] Following the abductive analytic approach we then returned to the SDM literature to understand how existing conceptualisations of sharing responsibility related to our findings and found that a continuum from what we came to conceptualise as *transactional* to interdependent *approaches* (box 1) helped our interpretations.[8 11]

---

**Box 1    Transactional and interdependent approaches to sharing responsibility**

**The transactional approach**
► Following this approach responsibility resides with one or the other actor at any given moment. The doctor and patient are considered autonomous and assumed to hold equal status in the relationship. The practice of sharing responsibility involves passing the responsibility from one to the other.

**The interdependent approach**
► The *interdependent* approach involves recognising how each party is reliant on the other. The doctor and patient are acting together to reach an understanding. Responsibility can reside with both the doctor and the patient at the same time, but this sharing may be asymmetrical in terms of experience, understanding or capacity to act.

---

**Table 1** Participant characteristics

| GPs (n=25) | | Patients (n=23) | |
|---|---|---|---|
| Female | 9 | Female | 12 |
| Age | | Age | |
| 34–38 | 7 | 26–45 | 5 |
| 39–43 | 2 | 46–65 | 7 |
| 44–48 | 5 | 66–85 | 10 |
| 49–53 | 6 | Not stated | 1 |
| 54–59 | 5 | | |
| Years as a qualified GP | | Outcome | |
| 0–9 | 9 | Cancer diagnosed | 5 |
| 10–19 | 7 | Cancer ruled out | 18 |
| 20–29 | 9 | | |
| Ethnicity white British | 21 | Ethnicity white British | 20 |
| Recruited via | | Recruited via | |
| TVCRN | 10 | GP | 18 |
| CCG | 7 | Other | 5 |
| Direct invitation | 8 | | |

CCG, Clinical Commissioning Group; TVCRN, Thames Valley Clinical Research Network.

### Patient and public involvement

This project was funded to understand GPs' and patients' experiences of safety netting in relation to cancer in primary care. In preparation we conducted a secondary analysis of patient interview data on the process of bowel and lung cancer diagnosis in England, Sweden and Denmark (NAEDI 2015 C7663/A17663) to identify safety netting issues for inclusion in the interview topic guide. Twenty-three adult patients were interviewed as part of this study: when their responses uncovered additional relevant themes these were incorporated into the flexible topic guide by the lead researcher. The main study findings were shared with all participants via a website summary of findings.

### RESULTS

In-depth, face-to-face interviews were conducted with 25 qualified GPs and 23 adult patients between 16 November 2016 and 14 June 2017 (table 1). We explore our findings according to the different ways in which responsibility was described, illustrated with direct quotes from the interviews, presented as predominantly *transactional or interdependent* approaches to acknowledge that descriptions can include characteristics of both. Hesitations and repetitions have been removed to aid readability.

### Transactional approaches to sharing the responsibility

After assessing a patient's symptoms and deciding a plan of action, GPs often reported aiming to 'hand over' the responsibility for their plan to patients who they assessed to have sufficient capacity to advocate for themselves. These patients would be expected to attend healthcare appointments, phone to obtain (normal) test results and return to the GP if symptoms persisted. 'If there's someone who seems to have full capacity to, and life is not too chaotic, then I think you hand over responsibility when you explain to them. You know, I think your duty is to inform and advise what their specific action should be.' [GP16, F (female), aged 30–39y]. This is consistent with transactional understanding where responsibility is passed from one 'actor' to another.

Patients recognised that not only was looking after their health morally responsible—'It's your health, it's your body and it's important you take care of it' [P23, F, aged 50-59y]—but also that GPs had limited time for chasing people up. 'If they've asked you and you don't go, there is a chance that it will be missed isn't it? So I think it's not just the doctors, it's up to the patients as well to do as they've been asked to. Otherwise they can't expect the doctors to remind them then, they're far too busy aren't they?' [P01, F, aged 70-79y]

One GP, who also reflected on the medico-legal importance of clearly documenting the advice given to the patient, concluded that there were limits to the responsibility that the GP should hold: 'I think ultimately if the person has capacity then it's their choice whether they come back or not. I can tell them everything that I think they need to do.[…] But if they choose not to come back I can't force them to. […] So at the end of the day I think it's up to the patient. And if they have an inoperable cancer because they've delayed, I won't feel bad about it if I've done everything I can to bring them back' [GP08, M, aged 50 years]

### Timeframes

GPs reported that they often suggested a timeframe for when to re-consult, and considered it a relatively reliable way of encouraging patients to return. GPs described various rules of thumb for the timeframes, for example GP20 allowed six weeks from the onset of back pain with no other 'red-flag' symptoms, saying: 'If they've come in after two weeks, I'd be saying, "four weeks". And there's no great science behind that, it's just a rough sort of guide.' [GP20, M (male), aged 50-59y]

No patients mentioned being involved in agreeing the timeframe for follow-up, although some indicated a willingness to be guided by the GP: 'He said, "What we'll do, we'll leave it for three weeks and let's see what happens". He said, "If it is food related it is highly likely it will sort itself out during that timeframe". Okay, so that's what we chose to do, made an appointment for three weeks hence.' [P12, M, aged 60-69y]

## Action plans

Communicating action plans and suggesting timeframes were intended to increase the likelihood of patients accepting the transaction of responsibility and acting accordingly. Patients appeared to expect and accept this approach, although some emphasised that even if information about symptoms had been given and arrangements for follow-up made, the GP still had a responsibility for their care.

According to GPs and patients, proposed action plans were usually communicated verbally but if the GP thought the patient was likely to forget they might put it in writing. 'I can think of a couple of patients I've given it to them there and then. I've written down what the plans are. "Do these bloods", and "You should hear from somebody in X, Y or Z weeks, and if not…", so that even if they can't do it, if they've got a carer they can show it to or a family member they can find out what's happening.' [GP09, M, aged 50-59y]. GP19 reported routinely providing written action plans, because he felt it was the best way to communicate and it also provided a practice record of what he had told the patient.

## Lack of contingency

It was often difficult for GPs to know where to draw the line between offering reassurance and maintaining responsibility for follow-up. However, patients may not feel able to re-consult with ongoing symptoms after being reassured. GPs showed that when a disconnect occurred it could cast a long shadow on their own practice and sense of caution and concern. For example, GP04 described a case in which a patient did not re-consult, even though the symptoms were getting worse, for 6 months. The GP had not initially suspected cancer, arranged no investigations or follow-up, and did not explain the circumstances in which a review in clinic would be appropriate. 'I had a look and it just looked like a lump on the nose, it doesn't look like anything. And because he had seen me about it he ignored it then, and it grew and it grew and it grew, and when he finally came back six months later, had an urgent referral for this sort of enlarging tumour, and it killed him in the end.' [GP04, M aged 50-59y]

## Interdependent approaches to sharing the responsibility

At the other end of the continuum, patients often reported feeling reliant on the GP to provide them with sufficient explanation about their symptoms and the rationale for the plan of action to enable them to be proactive in taking responsibility for follow-up. 'In a lot of situations, yes it would be the person's responsibility to follow things up. But I think that has to be on the basis that they understand why they're following it up. So they have to be given that information and all the possible things, you know, if they're checking for something they need to be told what they're checking for and why that would be serious and why it's important that they come back.' [P18, F, aged 20-29y]

## Explaining thinking

GPs reported encouraging active patient participation in the consultation and the subsequent diagnostic process by explaining their own thought process and uncertainties about what might be causing the symptoms. 'It would be rare for me not to say what I'm thinking. And I think that it […] helps the patient to know that you're taking this seriously and […] that you've picked up on their concerns.' [GP09, M, aged 50-59y] Some GPs described how on occasion they might deliberately try to increase the patient's level of concern about their symptoms if they suspected an unworried patient might not take the responsibility for follow-up. In such cases, rather than being overly reassuring, the GP might stress the potential seriousness of the symptoms to raise the patient's level of concern to the point where they would take on responsibility for follow-up. While necessary, this could be uncomfortable: 'It's never nice to frighten people but I think under certain circumstances you probably have to, to a certain extent.' [GP04, M, aged 50-59y]

## Patients taking the initiative

Patients could feel that their GP or the health system had let them down if they had felt the need to attend for follow-up sooner than suggested or to insist on a referral. Often their understanding of what was expected to happen relied on the GP telling them. If there was a mismatch over the expected symptom trajectory then the patient's decision to attend follow-up earlier than advised depended on knowing what was expected (eg, that she should feel better within 5 days of this treatment). Some patients (like P03, earlier) described times when they felt that they had to act on their own initiative to persuade the GP to provide the appropriate care or an urgent referral. Examples included re-consulting about persistent or worsening symptoms either sooner than the GP had explained would be expected or where no timeframe or even any follow-up had been suggested. For instance, P13 re-consulted after 2 weeks instead of the suggested three because a perceived lump in her throat seemed to be worsening, and she was concerned. 'And I went to see a locum, who was a very, very nice lady, very pleasant, and she said, 'Oh I think it's probably nothing to worry about', I'll prescribe this and I'll prescribe that — Beconase and what have you — 'go away and I'll see you again in three weeks' time'. And I went back to see her actually in two weeks' time because it seemed to be getting worse.' [P13, F, age not stated]

The GP or patient may also discount the need for follow-up if cancer is not suspected during the consultation. P03 said that she had consulted with four GPs on six occasions over a 3 to 4-month period with different symptoms before being referred and diagnosed with lymphoma. She explained that her GPs didn't plan a follow-up or describe when to re-access care. As a result, when her symptoms continued to worsen and new ones developed, she felt isolated in her determination to establish their cause. 'They never really expected me to come

back. And like I said, I felt like I was the one who was being proactive. And sometimes I feel, had I not been… […] And it was only me who was feeling the urgency of something being done really, you know, like really pretty soon.' [P03, F, aged 40-49y]

## DISCUSSION

GPs and patients talked about how responsibility for safety netting and follow-up actions can move and be shared throughout the consultation. In the transactional approach responsibility was passed between GP to patient, patients expected and were willing to accept responsibility, as long as they felt they had received sufficient instruction from their GP. Patients described leaving the consultation after discussing the reasoning behind the plan of action, but action plans were not routinely put in a written form, except when the GP thought their patient was likely to forget. In some cases, a transactional approach was described as a means for GPs to withdraw (medico-legally) from responsibility if patients did not follow their advice.

Knowing the patient, tailoring information, constructing preferences, achieving consensus and promoting relational autonomy are facets of an interactional care.[11 30] Interdependent sharing develops this insight by recognising the strategies that GPs report using engender mutual needs, goals and understanding: explaining the rationale for their actions; explaining their uncertainty about the cause of symptoms; and stressing the potential seriousness of the situation to raise the patient's level of concern. In this regard, the relationship between the GP and patient remained asymmetrical with the GP sharing more (or less) responsibility for ensuring that the patient accepts responsibility for follow-up.

Patients holding sole responsibility could be put at risk if: no follow-up had been suggested; no timeframe had been proposed; they had inadequate information on which to base follow-up decisions; they had been falsely reassured by previous consultations or test results without contingency planning; or if GPs did not address their concerns at re-consultation.

### Strengths and limitations of this study

We believe this is the first study to explore GPs' and patients' understandings of the ways in which responsibility for safety netting is shared in the context of possible cancer. Although our study was limited to one English county, we achieved a varied sample of GPs and patients. Being self-selected, the GP participants may have felt a stronger commitment to safety netting than others who declined to take part, and a wider geographical sample, including patients who lacked the capacity or willingness to take responsibility for follow-up, might have illuminated additional issues.

To minimise recall bias, interviews were conducted within 6 months of the safety netting episode described in the GP and patient accounts. As is always the case with reports, the participants may have forgotten, misunderstood or re-framed their experience depending on what happened next. For example, a patient who knows there was a delay in a cancer diagnosis may recall consultations with their GP differently from those who believed their diagnosis was prompt. Furthermore, people do not usually want to perceive, or describe themselves as, irresponsible; hence, they may report 'ideal' behaviours or present idealised versions of themselves to the researcher. However, we identified considerable variation in the reports and reflections of patients and GPs who have had recent experience of safety netting. These help us to understand, in the light of other literature, how the responsibility for safety netting is shared (or not) in everyday primary care.

Had we observed real-life consultations, this could have shown how safety netting is achieved in practice. Content and conversation analysis of video archives of routine GP consultations might offer an opportunity to study safety netting in context, but it remains to be established whether cancer relevant consultations are captured frequently enough for this to be an efficient design. Without interviews, the normative and value-laden meanings of responsibility that underpin how safety netting is understood would not have been available for analysis. The wording of the question about whose responsibility it was to make sure the patient returned for follow-up may have somewhat biased responses towards transactional approaches. We avoid reporting frequencies within the categories to reflect the nature of the data.

### Comparison with existing literature

Ideas of responsibility for safety netting in patient–GP relationships are derived from shared cultural norms that encourage or enforce certain behaviours. In her essay about how patients decide to consult a GP, Ziebland uses Robert Merton's concept of sociological ambivalence to propose five contradictory norms that govern citizen interactions with the healthcare system[31] including 'The good citizen trusts experts but recognises and accepts personal responsibility for own health.' and 'The good citizen accepts the doctor's reassurance about the low likelihood of a serious health problem but also listens to their own body and is prepared to challenge advice if the symptom persists or worsens' . Our findings provide further evidence of these contradictions which, without crystal clear, written, communication can lead to confusion about where the responsibility for follow-up is being held.

It is unsurprising that our interview data suggested asymmetry in the GP–patient relationship. Pilnick and Dingwall have shown that asymmetry in medical interactions has persisted despite decades of interventions aimed at increasing patient-centredness.[32] They argue that asymmetry is an inescapable function of the institution of medicine in society. The shift towards increased patient involvement has led to widespread use of a transactional style of shared responsibility which, while instructing

patients and relieving pressure on primary care, also brings risks to patient safety, especially in consultations where cancer is not initially suspected.

A study of cancer patients' experiences of the pre-diagnostic phase[26] showed that it was not unusual for English patients to leave the GP consultation unsure about what should happen next and under what conditions they should return; this was less common in a comparative sample of patients interviewed in Sweden. The authors concluded that clearly communicated action plans, which are a common feature of consultations in Sweden, should be used routinely in all consultations. Our findings align with this recommendation.

### Implications for research and practice

The unintended consequences of the ways responsibility is shared have implications for consultations with patients for symptoms that could represent underlying cancer. These implications are of broader relevance as symptoms of possible cancer are more commonly explained by a benign condition, by another serious disease or they resolve spontaneously. A clear explanation of the follow-up plan, including the underpinning rationale and ongoing uncertainties, is key to enabling patients to re-consult appropriately. Within this plan, a contingency should exist no matter what level of risk the GP perceives. This enables the patient to re-consult when symptoms persist after the GP had expected them to resolve. Without a contingency plan, patients who do not feel confident to take the responsibility represent a risk of loss to follow-up and, potentially, delayed cancer diagnosis.

Further research is necessary to understand the drivers of shared responsibility, including when it is used to distance the GP from the consequences of patients' actions. If, for example, increasing workload is a driver for the devolution of responsibility, then the healthcare system should take greater responsibility for safety netting through systems-based approaches to ensure safe follow-up and longer consultations. This should not just be for those patients the GP considers at highest risk and who comply with GP advice. If, on the other hand, GPs aim to enable their patients to make autonomous choices, more research is required to understand the most effective ways to communicate safety netting messages with patients. To achieve this, further research might analyse video recordings of GP consultations to examine how action plans are communicated and responsibility negotiated within and outside the consultation. It is not clear how widespread use of action plans might affect consultation rates. Reminders or action plans (written where necessary) used at the end of every consultation for symptom follow-up (not just those where the GP suspects cancer) would aid clarity and locate the 'Goldilocks' zone within which a return consultation is preferred.[31]

**Acknowledgements** We are grateful to the Oxfordshire Clinical Commissioning Group for supporting the study and helping to recruit General Practitioner participants. We also acknowledge the support of the National Institute for Health Research, through the Thames Valley and South Midlands Clinical Research Network, who helped recruit GPs into the study. Sue Ziebland is an National Institute of Health Research Senior Investigator.

**Contributors** BDN, JE, SZ and CB conceived and designed the study. BDN is principal investigator for the study and had oversight, JE was project lead. JE conducted the interviews. JE and JIM led the data analysis. Data on the rich theme of 'responsibility' were also examined by members of the research team (JE, JIM, CB, BDN, SZ). All authors were involved in drafting and commenting on the paper and have approved the final version.

**Funding** This study was funded by a Cancer Research UK project grant through their Early Diagnosis Advisory Group (Award number C50916/A21500).

**Competing interests** None declared.

**Patient consent for publication** Not required.

**Ethics approval** The study was approved by the South East Coast—Brighton & Sussex Research Ethics Committee (ref 16/LO/1468).

**Provenance and peer review** Not commissioned; externally peer reviewed.

**Data availability statement** De-identified transcripts from the study are archived at the Nuffield Department of Primary Care Health Sciences, University of Oxford until 31 October 2027 with participants permission, and are available under data sharing agreement to University of Oxford researchers for further analysis.

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
