## [Reviewer comments · BMJ Open]

ARTICLE DETAILS

TITLE (PROVISIONAL)	How do GPs and patients share the responsibility for cancer safety netting follow-up actions? A qualitative interview study of GPs and patients in Oxfordshire, UK.
AUTHORS	Evans, Julie; Macartney, John; Bankhead, Clare; Albury, Charlotte; Jones, Daniel; Ziebland, Sue; Nicholson, Brian

VERSION 1 – REVIEW

REVIEWER	Peter Lucassen Radboudumc Department of Primary and Community Care Nijmegen The Netherlands
REVIEW RETURNED	12-Feb-2019

GENERAL COMMENTS	Thank you for having the opportunity to review this qualitative study among GPs and patients about the important topic of safety-netting for cancer. The topic is relevant and has been addressed sparsely in the scientific literature. The study is well-designed and conducted with a sufficient number of participants. The composition of the research group seems adequate to me. However, I have some remarks: 1. When reading the Abstract for the first time I could not immediately grasp the difference between transactional sharing and dependent sharing. Reading further some ambiguity remained. In the analysis section on page 5, the authors distinguish transactional and dependent sharing and the refer to references 8 and 11; in both references the phrase 'dependent sharing' is not used; in reference 11 the authors of that publication use the words 'transactional care' and 'interactional care'. In my opinion this is much clearer. Moreover, in transactional sharing there doesn't seem to be much sharing. I propose revision of the wording here.2. In the Methods section the authors do not describe whether they performed data gathering and analysis cyclical. This is important because it provides the opportunity to check whether certain topics are underexposed or to introduce new topics.3. The paragraph 'Patient and public involvement' on page 6 could easily be integrated in the paragraph 'Interviewing'.4. Concerning the analysis it is unclear to me if the authors analyzed the themes deductively or inductively. In the Methods they state that they coded the transcripts and constructed themes using constant comparison, that there was a rich theme of 'responsibility' (inductive), and that they then returned to the SDM literature with the themes of transactional and dependent sharing (deductive). In the Results they only present these two themes. We do not see anything of the results left out of the description. Moreover, we do not have any idea of the content of an interview
--

	guide. What about the rich results of the responsibility theme? So, the reader is not involved in the process of selecting of the themes and does not have an overview of the broadness of the results. 5. For me the results are difficult to follow. I recognize (as a GP) transactional sharing and dependent sharing (or rather interactional sharing). My questions are: do GPs have one style or do they switch between these styles? How do they choose to switch? For which patients do they use each style? What are the elements of each style and how do these styles differ/correspond? How typical are timeframe and proposal of action plan for transactional sharing? Do GPs check whether the patient agrees with transactional sharing? What do patients think about these two methods. Do they always prefer one style, or does it depend on the problem at stake? What are the patients' experiences and wishes concerning involvement in decisions for follow-up? What do GPs consider as sufficient capacity for doing transactional sharing (page 7, line 15)? Is there a difference according to GPs in the information they provide in each of the two methods? 6. I'm very curious about how GPs perform this kind of sharing for follow-up. What do they tell patients? For it is very difficult to reassure patients that nothing serious is wrong and at the same time having to tell them that they should come back when the symptom persists. What kind of difficulties do GPs experience here? Do they avoid reassuring patients in these circumstances? 7. The text on page 8 line 6-20 does not seem to fit with transactional sharing, particularly not with sharing. 8. In the section Limitations of sharing the authors discuss the issue in a case of a GP who did not suspect cancer and therefore arranged no investigations of follow-up. This seems to me not a limitation of sharing because the GP did not have something to share about follow-up (although later this appeared to be a faulty conviction). 9. In the Strengths and limitations section the authors state that real-life observations would have been a better method. I think they are right, but I do not think that this method would be inefficient. Probably there are videorecordings from other studies in the consulting room that are available for this subject In conclusion I think that the analysis of the interviews has not sufficient depth, but maybe there is depth but remains out of sight because of the way the results are described.
--	---

REVIEWER	Ruth Swann Cancer Research UK / Public Health England partnership, UK
REVIEW RETURNED	13-Mar-2019

GENERAL COMMENTS	This is paper provides valuable insights into further understanding safety netting and enhances the current literature in this field. The manuscript is well written, the text is clear and easy to follow, with the very helpful addition of Box 1 and the Table providing a good overview. There are some minor comments that could be addressed: Abstract It would be useful to have the time-frame included i.e. interviews were within six months of safety netting. Introduction
--

	Line 3 – ‘There is a growing literature examining why patients choose to consult their doctor with symptoms that could represent cancer’. Suggest to change the words ‘choose to’. Line 7 – Could safety netting also be used where the symptoms did not necessarily indicate serious disease but the GP still wanted to monitor the symptoms? Line 45 - This sentence raises a good point generally, however at the time point of the safety netting the people under investigation are not yet, or might not become, cancer patients. Line 60 – The aim in the introduction has an additional point - ‘how might practices of safety netting be improved?’ – which is not included in the abstract. Methods Page 5, Line 32 – Was consent informed? Page 5, Line 52 – It is mentioned here that sub-themes were identified but these are not mentioned again in the manuscript Page 6, Line 3 – Typo ‘GPs and patients experiences’ change to ‘GPs’ and patients’ experiences’. Results It would be good to have a sense of whether there was more of one type of sharing model over the other, either in the text or indicated in Table 1. Discussion Reflecting on reference 9 which is mentioned in the Introduction (line 35), could the authors provide any insights into how SDM could be translated into routine practice given the results they have found? As the majority of patients were not diagnosed with cancer in this study, the findings could be used to enhance the knowledge of other disease areas. Out of the scope for this paper but it would be interesting to see whether people’s retrospective perceptions of safety netting were different if the individual had been diagnosed with cancer compared to those where cancer had been ruled out. The COREQ criteria checklist is complete, however the gender of the researcher is not specified within the text, as described in point 4 of the checklist.
--	---

VERSION 1 – AUTHOR RESPONSE

Reviewer: 1

Reviewer Name: Peter Lucassen

Institution and Country: Radboudumc Department of Primary and Community Care Nijmegen
The Netherlands

Please state any competing interests or state ‘None declared’: none declared

Please leave your comments for the authors below

Thank you for having the opportunity to review this qualitative study among GPs and patients about the important topic of safety-netting for cancer. The topic is relevant and has been addressed sparsely in the scientific literature. The study is well-designed and conducted with a sufficient number of participants. The composition of the research group seems adequate to me.

***Author response: thanks to reviewer one for these positive comments.

However, I have some remarks:

1. When reading the Abstract for the first time I could not immediately grasp the difference between transactional sharing and dependent sharing. Reading further some ambiguity remained. In the analysis section on page 5, the authors distinguish transactional and dependent sharing and the refer to references 8 and 11; in both references the phrase 'dependent sharing' is not used; in reference 11 the authors of that publication use the words 'transactional care' and 'interactional care'. In my opinion this is much clearer. Moreover, in transactional sharing there doesn't seem to be much sharing. I propose revision of the wording here.

***Author response: we thank the reviewer for taking the time to consider the appropriateness of our terminology of "sharing" responsibility to capture what we have learnt from the interviewee's accounts. We originally decided that the transactional and interactional frames proposed by Epstein (reference 11) encompassed more than the sharing of information and responsibility within the GP consultation and so did not use it entirely. We have, after much further discussion, decided that we could be clearer about the meaning of the transactional and dependent sharing and so now we describe a continuum of transactional to interdependent approaches to sharing responsibility (Box 1).

2. In the Methods section the authors do not describe whether they performed data gathering and analysis cyclical. This is important because it provides the opportunity to check whether certain topics are underexposed or to introduce new topics.

***Author response: thanks - we have clarified this point by adding the second sentence of the following two: "In addition to eliciting their experiences and views of safety netting, all participants were asked for their opinion on whose responsibility it should be to make sure that patients returned for a follow-up visit. From this starting point, JE developed the 'responsibility' theme cycling between the conduct of interviews and preliminary analysis."

3. The paragraph 'Patient and public involvement' on page 6 could easily be integrated in the paragraph 'Interviewing'.

***Author response: we believe a standalone section on PPI is required by the journal.

4. Concerning the analysis it is unclear to me if the authors analyzed the themes deductively or inductively. In the Methods they state that they coded the transcripts and constructed themes using constant comparison, that there was a rich theme of 'responsibility' (inductive), and that they then returned to the SDM literature with the themes of transactional and dependent sharing (deductive). In the Results they only present these two themes. We do not see anything of the results left out of the description. Moreover, we do not have any idea of the content of an interview guide. What about the rich results of the responsibility theme? So, the reader is not involved in the process of selecting of the themes and does not have an overview of the broadness of the results.

***Author response: thank you –we have modified the abstract and methods to make our methodological approach to the analysis clearer to the reader. From the methods: "Data on the rich theme of 'responsibility' were examined by members of the research team (JE, JM, CB, BN, SZ) using the One Sheet of Paper (OSOP) method, a qualitative mind-mapping approach to analysis (29). Following the abductive analytic approach we then returned to the SDM literature to understand how existing conceptualisations of sharing responsibility related to our findings and found that a continuum from what we came to conceptualise as transactional to interdependent approaches (Box 1) helped our interpretations (8, 11)."

In the Results, we describe a continuum of transactional to (inter)dependent throughout the manuscript, rather than a binary classification as suggested. The results therefore report a range of examples of situations when the GP-patient interaction lies farther towards either end of the continuum.

5. For me the results are difficult to follow. I recognize (as a GP) transactional sharing and dependent sharing (or rather interdependent approach to sharing). My questions are:

***Author response: it is important and reassuring to us (as practicing clinicians and researchers) that these descriptions of sharing/care are familiar to a practicing GP. In order to make the results easier to follow, we have introduced additional subheadings.

It is also reassuring that the article conceptually provoked many questions. In response to these questions, we highlight direct quotes from the revised results section.

do GPs have one style or do they switch between these styles? How do they choose to switch? For which patients do they use each style?

***Author response: Transactional.... . These patients would be expected to attend healthcare appointments, phone to obtain (normal) test results, and return to the GP if symptoms persisted. 'If there's someone who seems to have full capacity to, and life is not too chaotic, then I think you hand over responsibility when you explain to them. You know, I think your duty is to inform and advise what their specific action should be.' [GP16, F (female), aged 30-39y]. This is consistent with a transactional understanding where responsibility is passed from one 'actor' to another.

Interdependent... "At the other end of the continuum to transactional care, patients often reported feeling reliant upon the GP to provide them with sufficient explanation about their symptoms and the rationale for the plan of action to enable them to be proactive in taking responsibility for follow-up. 'In a lot of situations, yes it would be the person's responsibility to follow things up. But I think that has to be on the basis that they understand why they're following it up. So they have to be given that information and all the possible things, you know, if they're checking for something they need to be told what they're checking for and why that would be serious and why it's important that they come back.' [P18, F, aged 20-29y]

What are the elements of each style and how do these styles differ/correspond? Is there a difference according to GPs in the information they provide in each of the two methods?

***Author response: please refer to the revised Box 1.

What do GPs consider as sufficient capacity for doing transactional sharing (page 7, line 15)?

***Author response: we consider capacity in greater detail in a previous paper (citation 25) but have added a qualifying statement "...who they assessed to have sufficient capacity to advocate for themselves"

How typical are timeframe and proposal of action plan for transactional sharing?

***Author response: the following section has the subheading "timeframe": "GPs reported that they often suggested a timeframe for when to re-consult, and considered it a relatively reliable way of encouraging patients to return. GPs described various rules of thumb for the timeframes, for example GP20 allowed six weeks from the onset of back pain with no other 'red-flag' symptoms, saying: 'If they've come in after two weeks, I'd be saying, "four weeks". And there's no great science behind that, it's just a rough sort of guide.' [GP20, M (male), aged 50-59y]

No patients mentioned being involved in agreeing the timeframe for follow-up, although some indicated a willingness to be guided by the GP: 'He said, "What we'll do, we'll leave it for three weeks and let's see what happens". He said, "If it is food related it is highly likely it will sort itself out during that timeframe". Okay, so that's what we chose to do, made an appointment for three weeks hence.' [P12, M, aged 60-69y]"

Do GPs check whether the patient agrees with transactional sharing?

***Author response: it is the interdependent approach that the GP and patient work to reach consensus on who has responsibility for follow-up. We have modified Box 1 to clarify and simplify the two extremes of the continuum.

What do patients think about these two methods. Do they always prefer one style, or does it depend on the problem at stake? What are the patients' experiences and wishes concerning involvement in decisions for follow-up?

***Author response: Transactional... "Patients also recognised that, not only was looking after their health morally responsible — 'It's your health, it's your body and it's important you take care of it' [P23, F, aged 50-59y] — but also that GPs had limited time for chasing people up. 'If they've asked you and you don't go, there is a chance that it will be missed isn't it? So I think it's not just the doctors, it's up to the patients as well to do as they've been asked to. Otherwise they can't expect the doctors to remind them then, they're far too busy aren't they?' [P01, F, aged 70-79y]"

Interdependent 'In a lot of situations, yes it would be the person's responsibility to follow things up. But I think that has to be on the basis that they understand why they're following it up. So they have to be given that information and all the possible things, you know, if they're checking for something they need to be told what they're checking for and why that would be serious and why it's important that they come back.' [P18, F, aged 20-29y]

6. I'm very curious about how GPs perform this kind of sharing for follow-up. What do they tell patients? For it is very difficult to reassure patients that nothing serious is wrong and at the same time having to tell them that they should come back when the symptom persists. What kind of difficulties do GPs experience here? Do they avoid reassuring patients in these circumstances?

***Author response: we presume that this comment refers to the paragraph now entitled "lack of contingency". This paragraph highlights the importance of contingency planning even when there is 'certainty' in the GPs mind over the diagnosis. We have previously reported that GPs tend to offer safety netting advice when they perceive that the risk is high. This leaves patients at risk when the GP is incorrect or when there is a lower risk (but not no risk) scenario. We have modified the implications for future research paragraph in the discussion to elaborate on this point.

7. The text on page 8 line 6-20 does not seem to fit with transactional sharing, particularly not with sharing.

***Author response: after amending the terminology to 'care' rather than 'sharing', in response to point 1 above, we hope that this section is more appropriately framed. An action plan is a method used by some GPs to transfer responsibility to the patient. In this respect, actions represent a transaction of care between the GP and the patient.

8. In the section Limitations of sharing the authors discuss the issue in a case of a GP who did not suspect cancer and therefore arranged no investigations of follow-up. This seems to me not a limitation of sharing because the GP did not have something to share about follow-up (although later this appeared to be a faulty conviction).

***Author response: we agree this example does not fit into the narrative and we have removed it from the manuscript.

9. In the Strengths and limitations section the authors state that real-life observations would have been a better method. I think they are right, but I do not think that this method would be inefficient. Probably there are video recordings from other studies in the consulting room that are available for this subject

***Author response: we have edited our comment about video recordings to read "Had we observed real-life consultations, this could have shown how safety netting is achieved in practice. Content and conversation analysis of video archives of routine GP consultations might offer an opportunity to study safety netting in context but it remains to be established whether cancer relevant consultations are frequent enough for this to be an efficient design".

In conclusion I think that the analysis of the interviews has not sufficient depth, but maybe there is depth but remains out of sight because of the way the results are described.

***Author response: this conclusion is at odds with reviewer 2 who considers the manuscript to be “well written, the text is clear and easy to follow”. We hope that by addressing the questions posed we have made the manuscript clearer.

Reviewer: 2

Reviewer Name: Ruth Swann

Institution and Country: Cancer Research UK / Public Health England partnership,
UK

Please state any competing interests or state ‘None declared’: None declared

Please leave your comments for the authors below

This paper provides valuable insights into further understanding safety netting and enhances the current literature in this field. The manuscript is well written, the text is clear and easy to follow, with the very helpful addition of Box 1 and the Table providing a good overview.

****Author response: thanks for these positive comments.

There are some minor comments that could be addressed:

Abstract

It would be useful to have the time-frame included i.e. interviews were within six months of safety netting.

****Author response: we have edited the objective to read “To explore patients’ and GPs’ accounts of how responsibility for follow-up was perceived and shared in their experiences of cancer safety netting occurring within the past 6 months.”

Introduction

Line 3 – ‘There is a growing literature examining why patients choose to consult their doctor with symptoms that could represent cancer’. Suggest to change the words ‘choose to’.

****Author response: thanks we have removed the words ‘choose to’

Line 7 – Could safety netting also be used where the symptoms did not necessarily indicate serious disease but the GP still wanted to monitor the symptoms?

****Author response: yes and absolutely! We have changed this sentence to now read: “Safety netting is a strategy used to ensure that patients presenting with symptoms or signs are monitored until they have resolved or an explanation or diagnosis reached“

Line 45 - This sentence raises a good point generally, however at the time point of the safety netting the people under investigation are not yet, or might not become, cancer patients.

****Author response: thanks. By saying “Little is known about how patients and GPs negotiate responsibility for safety netting when cancer is a possible diagnosis” we do not mean to say the patient is a cancer patient, only that there is a possibility that cancer could be the diagnosis.

Line 60 – The aim in the introduction has an additional point - ‘how might practices of safety netting be improved?’ – which is not included in the abstract.

****Author response: we have removed this clause as it adds little other than confusion!

Methods

Page 5, Line 32 – Was consent informed?

****Author response: thanks – yes we’ve added “informed” in the sentence highlighted. We have also added “Those who responded were sent an information pack.” After the information about patient recruitment not through the GP.

Page 5, Line 52 – It is mentioned here that sub-themes were identified but these are not mentioned again in the manuscript

****Author response: thanks – this is a description of the method – we have modified it to read “...a qualitative mind-mapping approach to analysis (28).”

Page 6, Line 3 – Typo ‘GPs and patients experiences’ change to ‘GPs’ and patients’ experiences’.

****Author response: thanks – changed.

Results

It would be good to have a sense of whether there was more of one type of sharing model over the other, either in the text or indicated in Table 1.

****Author response: thanks – in response to the points made by reviewer one we have edited the manuscript to reflect that clinicians used a variety of consultation styles along a continuum from transactional to interdependent rather than there being one approach used above another.

Discussion

Reflecting on reference 9 which is mentioned in the Introduction (line 35), could the authors provide any insights into how SDM could be translated into routine practice given the results they have found?

****Author response: thanks – we feel that the edited manuscript demonstrates that a varied approach to SDM is employed in response to symptoms that could represent underlying cancer. In this regard, we hope we have demonstrated when SDM is successfully implemented in practice and the problems created when it isn't.

As the majority of patients were not diagnosed with cancer in this study, the findings could be used to enhance the knowledge of other disease areas.

****Author response: thanks – we have incorporated this point into the implications section: “The unintended consequences of the ways responsibility is currently shared have implications for consultations with patients with symptoms that could represent underlying cancer. These implications are of broader relevance as symptoms of possible cancer are more commonly explained by an alternative benign condition, resolve spontaneously, or less commonly by another serious disease.”

Out of the scope for this paper but it would be interesting to see whether people's retrospective perceptions of safety netting were different if the individual had been diagnosed with cancer compared to those where cancer had been ruled out.

****Author response: thanks and we agree – we have added this consideration to the strengths and limitations section. “As is always the case with reports, the participants may have forgotten, misunderstood or re-framed their experience depending on what happened next. For example, a patient who is aware they have experienced a delay in a cancer diagnosis may recall interactions with their GP differently to a patient who has not.”

The COREQ criteria checklist is complete, however the gender of the researcher is not specified within the text, as described in point 4 of the checklist.

***Author response: thanks – we have added that JE is female.

VERSION 2 – REVIEW

REVIEWER	Peter Lucassen Radboudumc, Department of Primary and Community Care
REVIEW RETURNED	27-Jun-2019
GENERAL COMMENTS	I'm very satisfied with the authors' responses of the points that I have raised in the first review of this manuscript. It is an important topic and a well-designed study.

	1. The manuscript now reads much easier for me now and I appreciate the change in wording and the description in the new version. 2. Thank you for clarifying the point of the cyclical process of data gathering and analysis. 3. OK, then I do not comment on that. 4. OK, I'm also satisfied with the new description in which you mention the abductive approach. The Results section reads much easier now and as a GP I think it is now more relevant for GPs. 5. I'm very happy with the elaborate response and the corresponding changes in the new manuscript. 6. The same goes for this point. 7. I agree with the authors that the text is now more appropriately framed. 8. Thank you for removing this section. 9. Thank you for the additional description about content and conversation analysis. In conclusion I think that the authors have responded to my questions very appropriately.
--	---

REVIEWER	Ruth Swann Cancer Research UK / Public Health England partnership, UK
REVIEW RETURNED	05-Jul-2019

GENERAL COMMENTS	The revisions have made the paper clearer especially with the re-classification of the term dependent to interdependent. I have some minor comments on the article which are included below: Page 2, Line 17: There is a typo, 'intedependent' has been written instead of interdependent. Methods There is repetition of the sentence on page 5 line 60 (last sentence of paragraph 'Analysis') 'A summary of the main study findings was shared....' With the sentence on page 6 line 13 (last line of the paragraph ' Patient and Public Involvement'). 'The main study findings were shared...' Results The results under the 'transactional approaches to sharing the responsibility' section are very clear, well defined and have a natural direction. The results section on interdependent sharing do take slightly more reading to understand how these are different to the transactional approach. Some of the examples given in this section could perhaps be applied to either approach. For example, patients in a transactional approach would also need to have a sufficient explanation about their symptoms and a rationale for the plan of action. The first three paragraphs under the transactional heading talk about responsibility being passed from the GP to the patient, and the patient accepting responsibility and understanding their role in this. The last sentence of the second paragraph on page 7 line 37 ('As one patient said....') talks about a shared responsibility between the GP and the patient. This raises an interesting point but it doesn't seem to fit within these paragraphs. This sentence is making a point about sharing responsibility where as the other paragraphs focus on passing responsibility to the patient.
--

	The paragraph starting on page 9, line 43 highlights a separate point to the heading 'explaining thinking'. This seems to be more of a lack of contingency point and could perhaps have it's own heading. Box 1: This is really useful to understand the terminology. The definition of transactional approach here describes turn-taking, however the text in the article focuses more on responsibility being passed from GP to patient but not particularly in turn-taking.
--	--

VERSION 2 – AUTHOR RESPONSE

Reviewer: 1

Reviewer Name: Peter Lucassen

Institution and Country: Radboudumc, Department of Primary and Community Care

Please state any competing interests or state 'None declared': None declared

Please leave your comments for the authors below

I'm very satisfied with the authors' responses of the points that I have raised in the first review of this manuscript. It is an important topic and a well-designed study.

1. The manuscript now reads much easier for me now and I appreciate the change in wording and the description in the new version.
2. Thank you for clarifying the point of the cyclical process of data gathering and analysis.
3. OK, then I do not comment on that.
4. OK, I'm also satisfied with the new description in which you mention the abductive approach. The Results section reads much easier now and as a GP I think it is now more relevant for GPs.
5. I'm very happy with the elaborate response and the corresponding changes in the new manuscript.
6. The same goes for this point.
7. I agree with the authors that the text is now more appropriately framed.
8. Thank you for removing this section.
9. Thank you for the additional description about content and conversation analysis.

In conclusion I think that the authors have responded to my questions very appropriately.

AUTHOR RESPONSE – we are very pleased the reviewer agrees with our revisions.

Reviewer: 2

Reviewer Name: Ruth Swann

Institution and Country:

Cancer Research UK / Public Health England partnership,

UK

Please state any competing interests or state 'None declared': None declared

Please leave your comments for the authors below

The revisions have made the paper clearer especially with the re-classification of the term dependent to interdependent. I have some minor comments on the article which are included below:

Page 2, Line 17: There is a typo, 'intedependent' has been written instead of interdependent.

AUTHOR RESPONSE – thanks.

Methods

There is repetition of the sentence on page 5 line 60 (last sentence of paragraph 'Analysis') 'A summary of the main study findings was shared....' With the sentence on page 6 line 13 (last line of the paragraph ' Patient and Public Involvement'). 'The main study findings were shared...'

AUTHOR RESPONSE – thanks, we have removed the former.

Results

The results under the 'transactional approaches to sharing the responsibility' section are very clear, well defined and have a natural direction. The results section on interdependent sharing do take slightly more reading to understand how these are different to the transactional approach. Some of the examples given in this section could perhaps be applied to either approach. For example, patients in a transactional approach would also need to have a sufficient explanation about their symptoms and a rationale for the plan of action.

AUTHOR RESPONSE – thanks, we have re-emphaised that this is not an either/or situation by lengthening the intro to the results to include "We explore our findings according to the different ways in which responsibility was described, illustrated with direct quotes from the interviews, presented as predominantly transactional or interdependent approaches to acknowledge that descriptions can include characteristics of both"

The first three paragraphs under the transactional heading talk about responsibility being passed from the GP to the patient, and the patient accepting responsibility and understanding their role in this. The last sentence of the second paragraph on page 7 line 37 ('As one patient said....') talks about a shared responsibility between the GP and the patient. This raises an interesting point but it doesn't

seem to fit within these paragraphs. This sentence is making a point about sharing responsibility where as the other paragraphs focus on passing responsibility to the patient.

AUTHOR RESPONSE – thanks, we have removed this quote.

The paragraph starting on page 9, line 43 highlights a separate point to the heading 'explaining thinking'. This seems to be more of a lack of contingency point and could perhaps have it's own heading.

AUTHOR RESPONSE – thanks, we have moved this quote into the “patient taking the initiative section”.

Box 1: This is really useful to understand the terminology. The definition of transactional approach here describes turn-taking, however the text in the article focuses more on responsibility being passed from GP to patient but not particularly in turn-taking.

AUTHOR RESPONSE – Thanks - we've modified that sentence to read “The practice of sharing responsibility involves passing the responsibility from one to the other.”